# Regional Electric Vehicle Fast Charging Network Design Using Common Public Data

**Nathaniel S. Pearre** [1,*], **Lukas G. Swan** [1,*], **Erin Burbidge** [2], **Sarah Balloch** [2], **Logan Horrocks** [2], **Brendan Piper** [2] **and Julia Anctil** [2]

1   Renewable Energy Storage Laboratory, Dalhousie University, 5217 Morris Street, 4th Floor, P.O. Box 15000, Halifax, NS B3H 4R2, Canada
2   Clean Foundation, 126 Portland Street, Dartmouth, NS B2Y 1H8, Canada
*   Correspondence: nathaniel.pearre@dal.ca (N.S.P.); lukas.swan@dal.ca (L.G.S.)

**Abstract:** Electric vehicles rely on public fast charging when traveling outside a single charge range. Networks of fast charging hubs are a preferred solution, but should be deployed according to a design that avoids both redundant infrastructure representing overinvestment, and "charging deserts" which limit travel by EVs and thus inhibit EV adoption. We present a two-stage design strategy for a network of charging hubs relying on common public data including maps of roadways and electrical systems, and ubiquitous and readily accessible daily traffic volume data. First, the network design is based on the electrical distribution system, roadways, and a target inter-hub driving distance. Second, the number of fast chargers necessary at each hub to support expected vehicle kilometers is determined such that queuing to charge is infrequent. A case study to prepare Nova Scotia, Canada for the 2030 electric fleet of 15% of vehicles results in a network design with an average hub catchment area of 1230 km$^2$ and 354 electric vehicles per fast charger, and ensures that they are equitably distributed and can enable travel by EV throughout the jurisdiction.

**Keywords:** electric vehicle; charging; network; infrastructure; capacity

## 1. Introduction

As part of achieving greenhouse gas reduction goals, many jurisdictions have adopted aggressive plans to transition the personal vehicle fleet from fossil fueled vehicles to zero emissions vehicles (ZEVs). Among ZEVs, Electric Vehicles (EVs; containing no fuel) account for 2/3rds of ZEV sales [1] and already achieve 20% or greater overall vehicle market share in China [2] and Europe [3]. In many areas, however, public fast charging infrastructure is lagging in both spatial extent, compromising transportation equity, and in capacity, resulting in frequent and lengthy queues to charge.

This paper presents a transparent, open source, data driven algorithm to aid in the design of a fully developed regional EV charging hub ecosystem for a target future year or EV population. The presented method accounts for highway infrastructure, electrical system constraints, and daily traffic flow volumes. If more precise and specific data are available, it can be simply extended to include seasonal traffic patterns and EV energy efficiency, regionally heterogeneity in population growth, EV adoption, and driving patterns. Importantly however, in the absence of detailed traffic studies, representative regional approximations can be used in concert with location specific daily traffic counts.

This method includes two separate steps: (i) the design of the charging hub network based on infrastructural features and EV fleet characteristics; and (ii) the design of each charging hub to satisfy the travel needs of EVs within its catchment area. The "catchment" of the hub is the geographic area across which this hub is conceptually responsible for providing enroute charging for EVs. In practical terms, it is generally the area within which a given hub is the closest hub, though traffic congestion or idiosyncratic roadways such as

routing around a lake or river, may make that inaccurate. These two steps will be treated sequentially in Sections 2 and 3, respectively.

Unlike many more data intensive and geographically limited strategies, only common daily traffic data are used, and no proprietary or 'black box' optimization routines are required. In contrast to previously published methods, no geographic constraints are placed on hub network development, nor is there a need for traffic or trip data more precise than the daily vehicle counts commonly collected by departments of transportation. We argue that in the development of regional infrastructure, in contrast to a geographically limited study, such precise and extensive Lagrangian (vehicle tracking) data requirements are an impediment, while Eulerian (location specific) datasets already exist.

### 1.1. Scope and Limitations

1.  **General site location**. This project seeks to identify potential sites for charging hubs by proximity to route junctions and electrical infrastructure. This analysis does not address precise charging hub placement, which will be subject to detailed investigation of road traffic planning, land use planning, land ownership/control, electrical system access, etc. It likewise does not address site design considerations such as charger/parking layout, pull-through stations for vehicles towing trailers, on-site renewable generation, integrated batteries, and associated facilities such as convenience stores or lounges, etc.

2.  **Passenger vehicles only**. Fleet and commercial vehicles, whether taxis, private hire vehicles, public transit, or commercial transportation have distinct needs [4] that ought to be assessed by the primary operators of these fleets.

3.  **Only EVs**. Many vehicle sales goals refer to ZEVs, which include plug-in hybrid electric vehicles (PHEVs) and hydrogen fuel cell vehicles (FCVs). Sales trends show that EVs are already a majority of ZEV product offerings, and it is assumed EVs will be the overwhelming majority of sales for the foreseeable future [1–3]. This is also the expectation in Germany [5]. Furthermore, neither PHEVs nor FCVs will impact the DCFC network capacity at the limit; Rather than queuing to recharge, PHEV and FCV drivers can resume travel more quickly by refueling.

4.  **Only DCFC**. The focus of this report is high power enroute charging at charging hubs. AC charging at home, work, or other frequented parking locations enables EV use [6,7], but does not displace DCFC needed for long distance travel. Electrified roads are also being proposed and investigated [8]. Such 'E-Roads' involve either conductive or inductive power transfer to moving vehicles to enable long distance travel [9]. If combined with smaller batteries E-Roads could yield systemwide cost savings [10], but these require significantly different siting considerations and are not addressed.

5.  **Enroute charging for long distance travel**. Commuting and routine daily driving, along with exceptional traffic surges such as those associated with large concerts or major sporting events are not explicitly addressed. Heavily trafficked commuter routes will require special consideration as detailed in Section 3.2.6. In most such cases, EV charging needs are better served by lower rate home/workplace/venue charging so drivers do not need to divert from their commutes [7,11]. Residences that lack access to home charging will primarily be served either by retrofitted on-site AC charging infrastructure, or supplemental DCFC charging hubs targeting such urban populations. These will face different siting considerations not addressed in this study's method [12]. The need for such charging may be large in areas where dedicated parking is scarce [13].

### 1.2. Case Study: Nova Scotia, Canada

To illustrate the method, a case study for the Canadian province of Nova Scotia is presented. Nova Scotia is a ~550 × ~170 km peninsula in eastern Canada, with a narrow isthmus and effectively one travel route connecting it to the rest of North America. A map

of the province, the principle '100-series' highways, major secondary routes, and existing DCFC sites differentiated by site power (greater than or less than 50 kW total at the site) is shown in Figure 1. Most sites house a single 50 kW DCFC. The location of Nova Scotia in eastern Canada is shown in the inset top left.

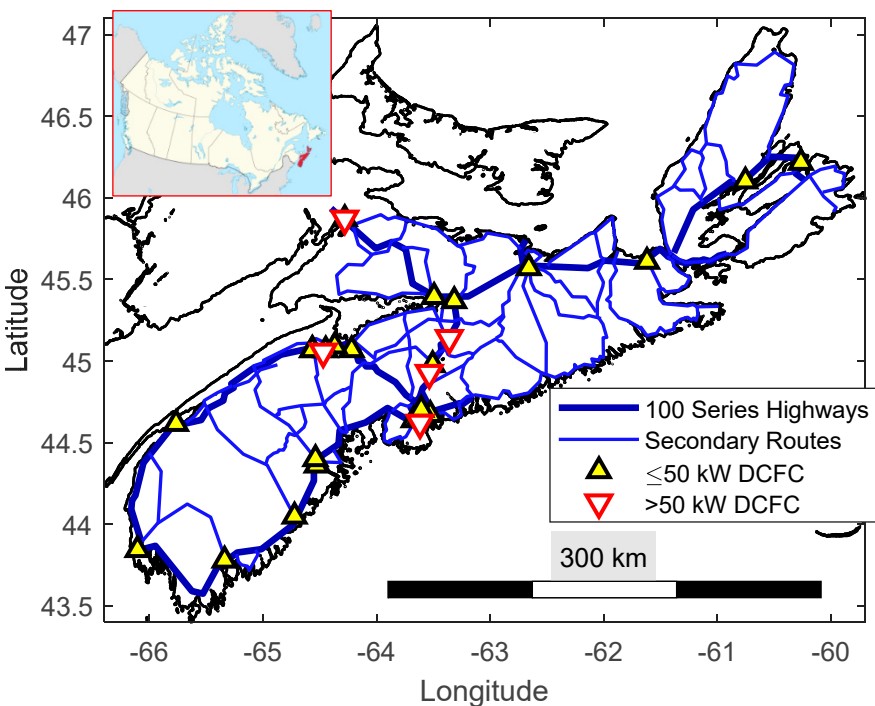

**Figure 1.** Map of Nova Scotia's primary ('100 series') highways and major secondary routes. The location of Nova Scotia in eastern Canada is shown for context (inset).

Nova Scotia has committed that by 2030 it will reduce its greenhouse gas (GHG) emissions by 53% compared to 2005 levels. The transportation sector accounts for 27% of the province's GHG emissions [14,15]. The provincial government has mandated that 30% of new light duty vehicle sales will be ZEVs by 2030 [16]. It is worth noting that the Canadian federal government has even more ambitious goals of 60% ZEV sales by 2030 and 100% by 2035. Based on these diverging policies and accounting for the rate of vehicle replacement, an EV fraction of the total personal vehicle fleet is calculated to be 15% in 2030, whichis used in the case study.

The existing DCFC network in NS (triangles in Figure 1) has benefitted from initial investments by government and private entities including the provincial electrical utility, Tesla Motors, and Petro-Canada. However, the existing infrastructure is inadequate, has reliability issues, frequent queues at DCFCs, and leaves many areas of the province inaccessible to EVs without careful planning. Without a strategic framework to address the need for robust and readily accessible DCFC infrastructure, the province will continue to experience access gaps. In addition, there is risk of investment in obsolete infrastructure sited in non-strategic locations. For example, current Government of Canada funding programs continue to allow the installation of 50 kW DCFCs, which are obsolete in terms of meeting the needs of newer, larger EVs with increased battery size and towing capabilities.

### 1.3. Litterature Review

Due in part to the ability to charge at home, EVs often do not need to detour to a charging station in day-to-day operations. Indeed, as EV ranges increase [17], modelling suggests less need to charge anywhere other than at home or work [18]. In aggregate, this will decrease the systemwide number of enroute fast charging hubs needed per EV, an effect reflected in at least some charging infrastructure models [19]. Notwithstanding

improving EV ranges, trips outside that range require the ability to recharge enroute, convenient to highways or other major travelways, reliably, and at high power. This need has been the focus of development and deployment of direct current fast charging (DCFC) infrastructure. The evolution of the market has led, according to a survey of Canadian drivers, to a decrease in concern over the abilities of EVs, and a corresponding increase in concern of the convenience, availability, and reliability of charging infrastructure [20].

A charging 'hub' is a site with multiple high power DCFCs at the same location. Charging hubs offer several benefits over sporadically distributed single or low-count charger locations, even if the same broad area density of DCFCs is achieved.

*For users:*

- There is one destination to drive to, where if one DCFC is occupied, another is nearby.
- Even if all DCFCs are occupied, wait time may be reduced by a single queue for whichever DCFC becomes available next.
- Multiple co-located DCFCs offer greater reliability through redundancy. The reliability risk of sites without redundant DCFCs is significant; A recent study in California's Bay Area found that roughly 1 in 4 DCFC were unable to charge a test vehicle [21].

*For builders/operators:*

- A charging hub may develop greater and more consistent traffic, so related business opportunities (convenience stores, fast food, etc.) have more potential market. This potentially presents both an improved experience for users and a local business opportunity.
- Having multiple DCFCs of the same make and model increase the efficiency of maintenance and repair, keeping of spare parts, manifesting in reducing cost and improving uptime.
- There are significant economies of scale in land acquisition, site preparation and permitting, as well as in burying electrical conduit, buying and placing transformers, etc. A large proportion of the cost of a DCFC charging hub in North America are 'soft costs', such as process costs and permitting [22].

Both Tesla [23] and Audi [24] seem to understand the mechanism of providing a charging ecosystem as a means to sell cars. Both are building charging hubs with multiple DCFCs at each, and increasingly an array of consumer amenities. Furthermore, the US Federal Government has proposed a design rule for charging infrastructure specifying four or more DCFCs at each site [25].

To avoid duplicated effort and to leverage economies of scale, central planning would seem to be a necessity for efficient investment. The design of such networks is described as "one of the most pressing challenges" for entities concerned with transportation infrastructure [26]. One such plan computes the total number of hubs and DCFCs needed in the USA, differentiating siting by 'cities', 'towns', and 'rural', but does not provide specific guidance on placement [19]. Other researchers have developed algorithmic charging hub location strategies. Some of these constrain their analyses geographically, such as in support of highway routes [27,28]. This leaves open the question of how a network of hubs should be designed to provide universal substitutability of EVs for fossil fueled vehicles.

Various regional network design strategies have been proposed, but there is no consensus even on what characteristics to prioritize. An agent based model focused only on travel patterns was used to support charging hub recommendations based on simulated charging choices during trips within the USA state of Washington [29]. Other researchers focus on locations with consumer amenities in the construction of an equation to score locations [30,31].

A key parameter in hub network design is the target driving distance between hubs, which must be related to the range of EVs [17]. EV range was found to account for a roughly six-fold variability in the necessary quantity of DCFC charging based on vehicle ranges [19]. Class leading EV ranges are trending up. At present, the longest-range EV available in Canada can travel more than 800 km between charges in formal drive cycle testing [15],

and concepts have been proposed with 1000 km ranges [32,33]. However, charging hub networks should be designed to accommodate vehicles with lower range, as well as those with their range reduced by towing and/or inclement weather.

There is little agreement in the literature about appropriate inter-hub spacing. A Canadian model uses a value of 65 km along a busy highway [27]; A 2022 proposal from the US White House proposed a nationwide network of charging hubs with a target spacing of 80 km [25]; A 2017 report by the US Department of Energy uses a target hub spacing of 112 km [19]; Tesla Motors, known for offering EVs with higher range than many of their competitors, install hubs with a maximum spacing of ~175 km, [23]; Finally, an infrastructure model for Western Australia uses a target hub spacing of 200 km [34].

## 2. Hub Location Method

The first phase of the design is laying out the network of charging hubs.

### 2.1. Hub-Eligible Locations

In order to make EVs a fully viable substitute for conventional vehicles, network design must insure equitable access to all drivers and all locations. This proposed method of full geographic coverage relies on three steps: (i) identify hub-eligible locations; (ii) seed and populate the network of 'rigorous' hub sites based on hub-eligible locations and the target hub spacing; and (iii) correct gaps in the network by adding 'loosened criteria' sites.

#### 2.1.1. Availability of Electrical Power

The electrical power system is made up of generation, transmission, distribution, and loads. Transmission line voltage is converted at substations to distribution line voltage (functionally, anything below about 35 kV [35]). Distribution lines are routed throughout communities to loads. A large charging hub may have instantaneous power draw of several MW to several tens of MW. Such a load could be quite significant compared to the distribution system capacity in many areas. In addition, all high power DCFC equipment requires 3-phase electricity as an input, while some distribution systems are single-phase (sometimes called split phase).

Without detailed and often proprietary data on the geographical layout of the distribution system, locations that are a long distance from 3-phase distribution substations are at risk of requiring significant additional expense to run new distribution lines to overcome limitations of inadequate conductor sizes, single phase feeders, or simply lack of electrical service. The first geospatial selection criterion is thus proximity to a three-phase electrical distribution substation. This criterion does not replace location specific grid impacts analysis and does not guarantee that new distribution lines will not be needed between the substation and any given location, but significantly reduces the maximum possible run of new line, and associated cost risk.

#### 2.1.2. Supporting Multiple Routes

The second key consideration for charging hub siting is enabling travel for as many drivers on as many different routes as possible. It is taken as a premise that travelers would not want to divert significantly from a preferred route between origin and destination points to access charging. Most drivers seeking charge will be on well traveled and high-capacity highways and routes, i.e., it is exceedingly uncommon to drive hundreds of kilometers without using major highways, even in remote areas. Travel between numerous and disparate origin-destination pairs will pass by the junctions of major routes. Where a major east–west route crosses a major north south route, there will be vehicles passing from, and bound for, each cardinal direction (e.g., from S to N, W, and E, from W to N, E, and S, etc.). That makes such crossroads efficient places for charging infrastructure.

It is worth noting that in the jurisdiction of the case study and elsewhere in North America, facilities within three kilometers of highway off-ramps can be advertised on the

highway in official signage [36]. Such areas are also likely to contain land zoned such that a charging hub is possible.

The map of three phase substations and major route junctures within Nova Scotia is shown in Figure 2.

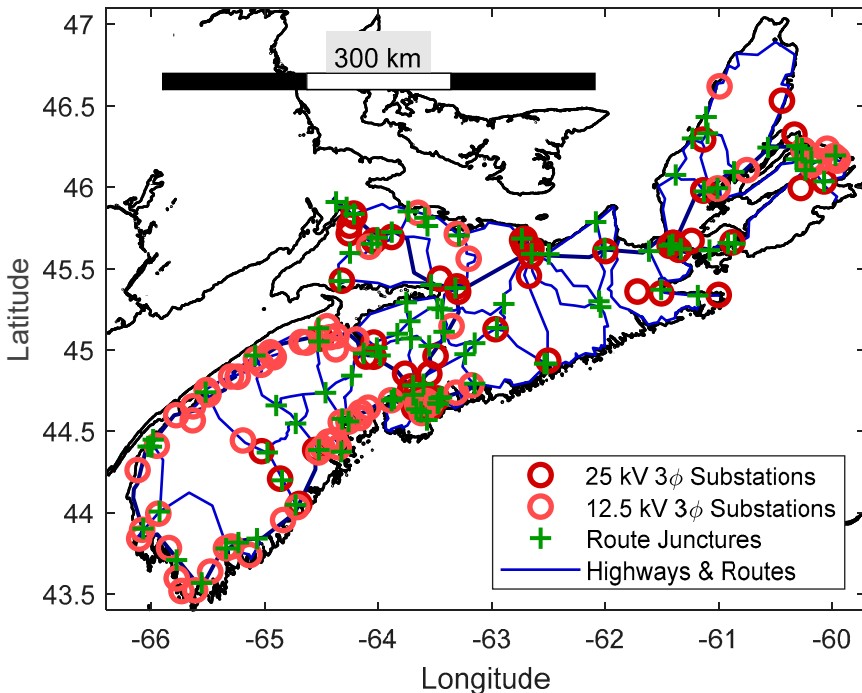

**Figure 2.** Map of major route junctures (green '+') and 3-phase distribution substations (red and pink circles).

## 2.2. Target Maximum Hub Spacing

Target maximum driving distance between charging hubs is the primary design variable in network layout. As discussed in the literature review, other authors and agencies have proposed target spacing between 65 km and 200 km. The benefit of increasing hub spacing is that fewer individual construction projects are necessary, and economies of scale suggest that total investment may be reduced. In counterpoint, the benefit of decreasing target hub spacing and increasing the number of hubs is that EV drivers on unusual routes, those who are 'surprised' by the need to charge enroute, or those who opt to bypass one charging hub for the next will not have as far to go to charge. Additionally, having 'the next hub' closer improves system reliability in the face of extreme events such as a power outage at one charging hub or pulses of traffic from major events.

With a high penetration of EVs, even a relatively short target spacing would in most instances require hubs that realize most of the benefits of economies of scale in construction. For the case study, a target hub spacing of 50 km was chosen. This value was chosen to support the travel of shorter range EVs in inclement conditions (e.g., in snow) or when heavily loaded (e.g., towing).

## 2.3. Hub-Eligible Location Down-Selection

The two geographically fixed location criteria, proximity to a substation and route junctions, populate a map with a selection of hub-eligible locations. Ideally, these points will be dense on length scales defined by the target hub spacing. In such areas the procedure used to establish 'rigorous' hub sites is summarized by the following:

1.  Start at **a 'seed' location** with multiple large or highly trafficked highway routes and abundant electrical infrastructure. In the case study, the junction of the Trans Canada

    highway with the highway heading to the provincial capital (indicated in Figure 3) was selected as a seed.

2.    Along each route to/from the seed location, **grow the network out from the seed** by identifying hub-eligible locations that are separated from the seed by roughly the target hub spacing (50 km). If multiple hub-eligible locations exist on the same route a suitable distance from the seed location, one must be selected. Secondary criteria for this selection may include accessibility (even within the context of all hub-eligible locations definitionally being near route junctions), amenities present, etc.

3.    Each selected location becomes a new seed.

4.    **Repeat** this 'network growth' from each new seed, until all major routes are covered to the extent permissible by hub-eligible locations.

5.    **Surround cities** with hubs at multiple hub-eligible locations. This will likely result in hub spacing well below the target hub spacing, but it reduces the risk of EV drivers visiting the city having to cross the city to charge before returning home.

*2.4. Charging Deserts and Loosened Criteria Hub Sites*

    In rural settings, the output of these steps may still leave large areas without access to charging hubs. These are termed "charging deserts" [37], and constitute both a loss of the generalizability of the method and, if left unaddressed, a loss of transportation equity. To address charging deserts, additional hub locations need to be selected, which requires loosening one or more of the criteria that define hub-eligible locations: proximity to substation, proximity to a route juncture, and distance from nearby hubs. Note that since consistent hub spacing is the driver of regional equity, it should be compromised only as a last resort.

    In the case of a charging desert where there is a convenient 3-phase substation but no route juncture, a smaller community or juncture of tertiary routes (not shown on any of these figures) may be suitable. In the case of a route juncture that is a suitable distance from other hubs, a greater distance from a substation may be acceptable, so long as 3-phase electrical supply is present (establishing this may require access to detailed data about the distribution system or may be ascertained by visual inspection in person or using Google Street View or similar tools). Such locations will likely be less trafficked hubs (fewer DCFCs), so peak loads will not be as high as at hubs along primary routes, and electrical infrastructure requirements may be reasonably relaxed.

    To produce the final hub layout for the case study shown in Figure 3, 35 'rigorous' hub sites within 3 km of route junctures and within 5 km of 3-phase substations were selected. These were augmented with 10 additional sites that required 'loosened criteria' but filled in charging deserts.

*2.5. Hub Network Design Evaluation*

    The 45 hub sites indicated in Figure 3 cover the 55,284 $km^2$ of provincial land area, giving an average hub catchment area of 1230 $km^2$. This area is equivalent to a square measuring 35 km $\times$ 35 km, or a 40 km diameter circle. The reduction of these geometric shape measurements relative to the target hub spacing likely relates to the proximity of many of the hub sites to the ocean, so much of their ideal geometric catchment is water.

    More precisely, the distance between sites shown in Figure 3 can be computed. The distribution of such distances (measured along roadways using Google Maps routing) is shown in Figure 4, along with the cumulative distribution function of that distribution.

    From Figure 4 it is evident that a hub spacing of 40–70 km is most common, as was the intent, though it is also obvious that a significant amount of variability exists about the 50 km target. However, about 80% of all inter-hub distances fall within the range of 30–70 km, and only 5% of inter-hub routes are more than 80 km. The longest inter-hub distance on the distribution is 117 km. That particular route could be replicated covering a total of ~140 km (20% more distance) to access either of two intermediate hub locations. The distribution refers to driving distance along all provincial numbered routes in each catchment.

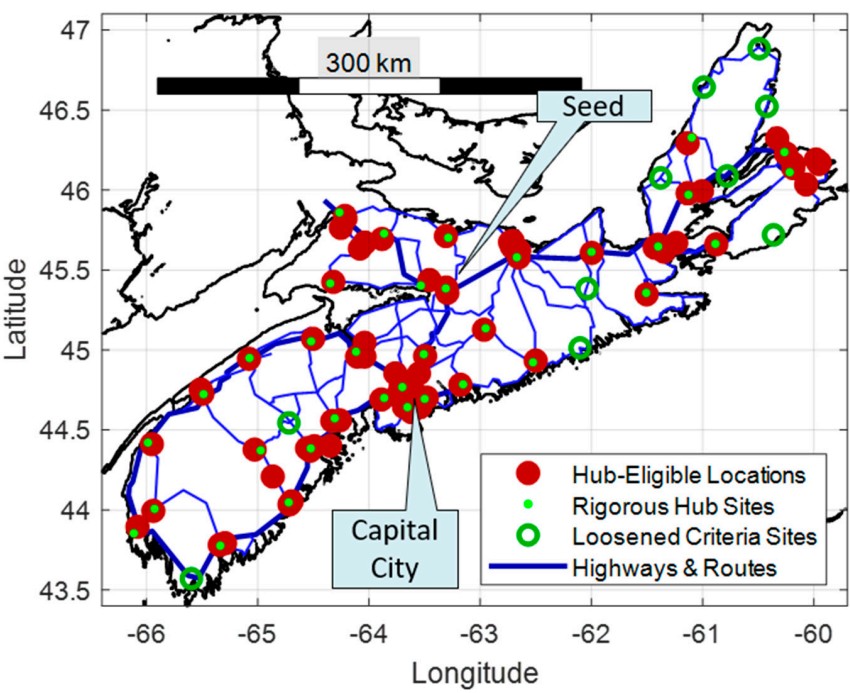

**Figure 3.** Hub-eligible locations (dark red dots) must be within 5 km of 3-phase distribution substations and within 3 km of route junctions. Rigorous sites (green dots) incorporate 50 km target hubspacing. Loosened criteria sites (green circles) are added to fill in charging deserts.

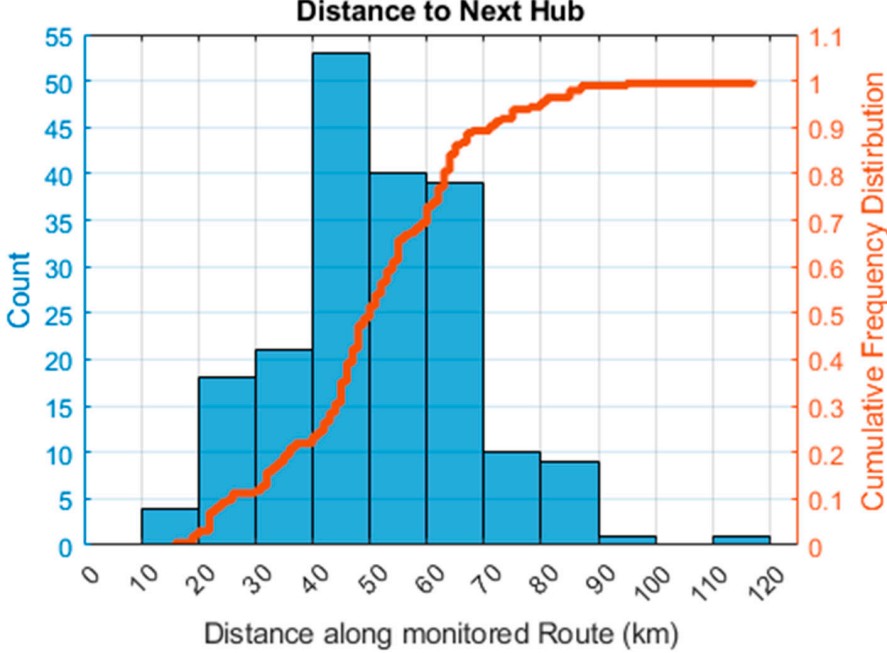

**Figure 4.** Distribution of inter-hub distances (on-road, not straight line).

## 3. Hub Sizing Method

Hub design must ensure adequate infrastructure at each site for both traffic volumes seeking charging and for reliability/redundancy.

### 3.1. DCFCs per Hub Equation

Once a hub network layout has been established, it is necessary to determine how many DCFCs are needed at each site to supply the energy needed by EVs driving within

each hub's catchment. The number of DCFCs needed at each hub is a function of six factors, many of which can vary by season, by hub site, or by route within a catchment.

Two factors may be tailored to the catchment of each charging hub:

- $Pop_Y/Pop\ EV_Y/Veh_Y$: The expected EV population in some target year *Y* relative to today's total traffic volumes. The subscripted '*Y*' indicates conditions in year *Y*, *EV*/*Veh* is the fraction of vehicles which are EVs, and *Pop* indicates (human) population;
- $P \times \Psi$: The total power *P* (kW) a single DCFC can provide through $\Psi$ cordsets;

Additionally, the product of four factors that may vary with individual routes must be summed across *M* routes in this hub's catchment:

- $\overline{C}$: Average electricity consumption (kWh/km); The average energy efficiency of the EV fleet;
- $UF$: Utilization Factor; How intensively a DCFC can be used;
- $N \times D$: Number of vehicles, Distance of route; The number of daily vehicles kilometers travelled on this route;
- $F_{DC}$: Fraction of charging at DCFC (%); The proportion of EV charging done enroute vs. at home or work.

These six components are combined according to Equation (1) (modified from [38]). Each component of Equation (1) is discussed as a subsection of Section 3.2.

$$\text{DCFCs at this hub} = \left( \frac{Pop_Y}{Pop} \times \frac{EV_Y}{Veh_Y} \right) \times \frac{1}{(P \times \Psi)} \times \sum\nolimits_{R=1}^{M} \frac{\overline{C}}{UF} \times \left( N \times \frac{D}{2} \right) \times \frac{1\,\text{day}}{24\,\text{h}} \times F_{DC} \qquad (1)$$

### 3.2. Application of Equation (1)

To illustrate the application of Equation (1), each term of the equation is discussed in greater detail below, and in the context of the case study. Depending on available data and regional traffic characteristics, it may be appropriate to compute Equation (1) twice, once for seasonal peak traffic volumes $N_{Max}$, and once for worst-case energy consumption $\overline{C}_{Max}$, retaining the larger of the two resulting values of 'DCFCs at this hub'.

### 3.2.1. Change in EV Population at Hub

The first terms in Equation (1), $Pop_Y/Pop \times EV_Y/Veh_Y$, is a multiplier for the expected EV traffic volume in a target year *Y*. $Pop_Y/Pop$ refers to overall population growth, and by inference traffic volume growth at this hub, relative to the year in which traffic volume data (*N*) are available, while $EVs_Y/Veh_Y$ refers to EV fleet fraction anticipated by year *Y*.

### 3.2.2. DCFC Power

$P \times \Psi$ is the peak power [kW$_{\text{potential}}$] each DCFC can supply. The number of cordsets per DCFC, $\Psi$, which has units [Vehicles/DCFC] has a default value of one (1), but may take on values other than one if the DCFC equipment can supply multiple vehicles at once., e.g., Tesla's 2nd generation superchargers supplied 150 kW to two heads [39], and ABB's Terra 360 supplies 360 kW to four heads [40]. In such a power sharing arrangement $P \times \Psi$ must remain the total power of the DCFC. Note that power sharing may increase the practicable utilization of a DCFC by avoiding the in-use charging power limitation imposed by EV battery voltage limitations. This is a matter that requires further study and an update to [38] and possibly increased values of *UF*.

The value of $P \times \Psi$ used in the case study was 350 kW for 'rigorous' hub sites, while as a blanket precaution, all 'loosened criteria' sites were assigned a *P* value of 150 kW to accommodate the possible limitations of the distribution system at some distance from an electrical substation.

### 3.2.3. Fleet Average Energy Consumption

The sum at the end of Equation (1) $\sum(\dots)$ refers to the set of *M* non-duplicative routes within this hub's catchment which must be evaluated separately. "Non-duplicative" in this

context means a driver transiting the catchment of this hub is unlikely to use more than two such routes, one arriving and one departing the catchment.

$\overline{C}$ is the weighted averaged energy consumption of EVs, in units of [kWh/km]. This value is a design decision informed by observed and published energy efficiencies of EVs [15]. Note that published values may be adjusted according to the characteristic of the route (highway vs. secondary road vs. urban) or the fleet mix [41], and additionally that $\overline{C}$ may vary throughout the year [42,43].

In the case study, no route-specific adjustments are used, but cold conditions are addressed by a uniform 25% energy consumption penalty. As EV designs develop better cabin and battery insulation and heat pumps, that will reduce winter consumption penalties. EV energy consumption of cars, trucks, and towing trucks should be considered. In the case study, mode split and vehicle efficiency values shown in Table 1 were adopted, guided by market data and trends [44,45].

**Table 1.** EV fleet breakdown and efficiencies.

| Type | Fleet Fraction | Nominal Efficiency | Winter Efficiency |
|---|---|---|---|
| Car | 25% | 175 Wh/km | 219 Wh/km |
| Truck/SUV | 73% | 300 Wh/km | 375 Wh/km |
| Passenger Vehicle Towing | 2% | 500 Wh/km | 625 Wh/km |
| Weighted Avg | | 273 Wh/km | 340 Wh/km |

### 3.2.4. DCFC Utilization

$UF$ is the unitless utilization factor [$kW_{Average}/kW_{Potential}$], defined by [38] as the total quantity of energy dispensed by a DCFC within a multi-day time window, divided by the theoretical maximum amount of energy that could be dispensed within that window. In effect it is the load factor of the DCFC. $UF$ is a design decision; High values of $UF$ require frequent queues to charge, while lower values require less queuing, but make the economics of DCFC site operation more challenging. To keep the probability of queuing to 10%, [38] suggests a maximum value of $UF$ of 7–17% depending on traffic patterns. Specifically (where data are available) if traffic is more evenly distributed throughout the day a higher $UF$ may be used, while on routes with more uneven traffic distribution, a lower $UF$ is suggested. Note that at low overall utilization, it may be important for charging hubs to have alternative revenue streams, as only selling electricity, a relatively low cost good, may be insufficient [46].

In the case study, a value of 10% $UF$ was used throughout.

### 3.2.5. Vehicle Kilometers in Catchment

In Equation (1) $N \times D/2$, in units of [vehicle kms/d] is the daily vehicle kilometers on this route in this catchment. The term $N$, with units of [vehicles/day], is the observed daily traffic volume along that route. Where seasonal data are available, the value of $N$ should account for seasonality of traffic volumes, i.e., one value for peak traffic volumes, and a second value for worst case energy consumption $\overline{C}$. The second term in the sum, $D$, with units [km/hub], is the distance to the next hub along this route. In idealized terms, $D$ is the target hub spacing, though geographic/infrastructural realities make perfectly regular hub spacing unlikely. Note that the target spacing $D$ is divided by two; Half of the distance along the route (and by inference half of the energy) is accounted for in the next hub's catchment.

Data for the case study were sourced from the Nova Scotia open data portal [47], and route specific values vary from $<1 \times 10^4$ to $>1 \times 10^6$ veh-km/d.

### 3.2.6. DCFC Charging Fraction

The final term in Equation (1), $F_{DC}$, in unitless [kWh$_{DCFC}$/kWh$_{Total}$], is the average fraction of energy sourced from enroute DCFCs along this route. It is recognized to be a function of EV range [17,19], but is also a function of geography, e.g., proximity to a major commuter destination [27]. Within the literature, $F_{DC}$ is subject to a range of values from 1.5% in an Austrian modelling study [46] to 5% in a USA modelling study [48], to 10% in a Canadian observational study [6].

Some other authors approach this variable from slightly different perspectives: A study focusing on Alberta and Ontario, Canada compute a roughly analogous 'Long-Distance Traffic Fraction', finding values of 2.5–80% depending on location and calculation method, methods used rely on less commonly available and more highly granular traffic data [27]. A study of driving patterns in Denmark found that public charging near residences and workplaces could reduce the need for other public charging by 87% among those without access to at-home charging [49], implying that residential and workplace charging makes up something like 87% of all charging, leaving 13% to enroute DCFC.

$F_{DC}$ will be strongly influenced by vehicle range, because longer range vehicles will need to visit enroute chargers less frequently [50]. The sensitivity of the need for public charging to this parameter is illustrated in a travel model [19], where changing the value [kWh$_{Home}$/kWh$_{Total}$] from 88% down to 82% increased the need for DCFC infrastructure by a factor of about 2.5 [19]. It is not explicit in their analysis how [kWh$_{DCFC}$/kWh$_{Total}$] changed in this sensitivity analysis, but one might presume it increased by 50%.

In the case study we adopt the recent findings of [6] and use 10% as the default fraction of energy EVs source from DCFCs for all proposed hub locations; hub or route specific values were not used.

### 3.3. Minimum DCFC Count

The output of Equation (1), in DCFCs per hub needed either based on peak driving season (maximum $N$ in Equation (1)) or minimum EV efficiency (maximum $\overline{C}$ in Equation (1)), can take on any positive value, depending on the traffic data for the selected hub location and other inputs. Fractional values must clearly be rounded up. In some areas with low traffic volumes, the output of Equation (1) will be a very small number, down to 0.6 DCFCs at one rural hub in the case study. This result is not a failure of the hub network design, since the goal of the network is to control inter-hub distances, and the network design does not discriminate against areas with low traffic volumes. Pursuant to the need for reliable hubs with redundant infrastructure, and similar to the recommendations of four (4) by [25], all hubs have been assigned a minimum of three (3) DCFCs. In addition to reliability, this will avoid the cost-inefficiency associated with single- and double- DCFC installations which will likely require upgrade as the EV fraction exceeds the 15% calculated for 2030.

## 4. Results and Discussion

The final design for hub locations and count of DCFCs per hub, determined for the case study of Nova Scotia, Canada in the year 2030 when 15% of vehicles are projected to be EVs, is shown in Figure 5.

In total, Figure 5 shows 45 hub locations housing a total of 318 DCFCs. This equates to 354 EVs per DCFC, which is a larger value that that found in [19], due perhaps in part to using higher power DCFCs. The largest site on the network is near the center of the capital city, projected to require 24 DCFC units each rated 350 kW by 2030. This may reflect an underrepresentation of the impact of commuter traffic on the site-specific values of $F_{DC}$ for a site on a major artery into the capital city.

The charging hub network shown in Figure 5 exhibits a range of DCFC counts, corresponding to a range of catchment areas and traffic volumes. In Table 2, each hub is listed by community name and sorted by average power delivered.

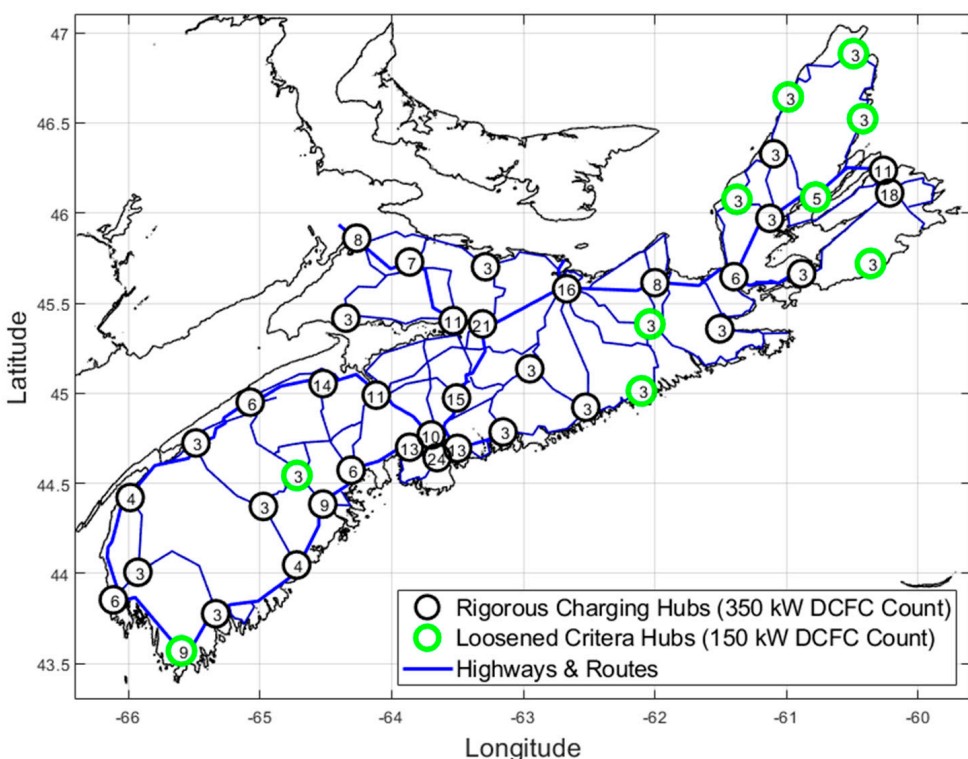

**Figure 5.** Charger counts at 350 kW (or 150 kW, green) each, at each hub location to support 15% EV fleet fraction.

For each hub in the proposed network, Table 2 indicates the DCFC count and the power rating of each. Most hubs use the default value of 350 kW, but 10 show the 150 kW rating assigned to 'loosened criteria' sites in green. Peak load at each hub is the product of the DCFC count and DCFC power at each site. Average load and monthly energy throughput (columns 5 and 6) are intermediate products from Equation (1). The monthly total electricity dispensed across the network is 6.7 GWh, representing roughly 10% of the total energy consumed by the fleet of EVs in the province.

The final column in Table 2 is the ratio of the average hub load to the peak power. It reaches a maximum value of 10%, reflecting the default value of *UF* (by design input), to keep the probability of queuing to about 10% [38]. Many hubs show less than 10% utilization because DCFC counts were rounded up and have a minimum value of three (3). These steps result in just under half (21 of 45) of the proposed charging hubs having the minimum allowed number of DCFCs, while 8 of 45 have utilization below 5%. Several factors contribute to this apparent over-investment:

1.  Emphasis was placed on access equity, resulting in hubs at locations with relatively little traffic.
2.  The high-power default specification of the DCFCs (350 kW), which increases the throughput of each unit, decreasing the number of DCFCs needed at each hub.
3.  The desire for hub reliability via DCFC redundancy; Placing a minimum of three (3) DCFCs at each hub, even if traffic volumes do not support it.

In this context, it is important to note that the case study was targeted to the year 2030, when 15% of the vehicle fleet is projected to be EVs. Assuming provincial or federal EV adoption targets are realized, less than 5 years later the EV fraction will have doubled to 30% and the need for the number of DCFCs will have doubled with it. Assuming regionally homogeneous traffic growth and EV fleets, most of the least-used hubs will cease to be under-utilized just a few years after this target date.

**Table 2.** Listing of charging hub with DCFC count, DCFC power, potential peak power draw, average electricity consumption, monthly electricity sales, and average utilization.

| Site Name | DCFCs | Power Each (kW) | Peak Hub Load (MW) | Hub Avg Load (kW) | Monthly Sales (MWh) | Utilization (%) |
|---|---|---|---|---|---|---|
| Clayton Park | 24 | 350 | 8.4 | 807 | 586 | 9.6% |
| Truro | 21 | 350 | 7.4 | 733 | 532 | 10.0% |
| Sydney Riv | 18 | 350 | 6.3 | 617 | 448 | 9.8% |
| New Glasgow | 16 | 350 | 5.6 | 533 | 387 | 9.5% |
| Elmsdale | 15 | 350 | 5.3 | 496 | 360 | 9.4% |
| N. Alton (Kentville) | 14 | 350 | 4.9 | 455 | 331 | 9.3% |
| Westphal | 13 | 350 | 4.6 | 455 | 330 | 10.0% |
| U. Tantallon | 13 | 350 | 4.6 | 424 | 308 | 9.3% |
| Glenholme | 11 | 350 | 3.9 | 377 | 274 | 9.8% |
| Windsor | 11 | 350 | 3.9 | 375 | 272 | 9.7% |
| Florence | 11 | 350 | 3.9 | 371 | 270 | 9.6% |
| Lower Sackville | 10 | 350 | 3.5 | 346 | 251 | 9.9% |
| Bridgewater | 9 | 350 | 3.2 | 311 | 226 | 9.9% |
| Aulac | 8 | 350 | 2.8 | 278 | 202 | 9.9% |
| Antigonish | 8 | 350 | 2.8 | 267 | 194 | 9.5% |
| Oxford | 7 | 350 | 2.5 | 228 | 166 | 9.3% |
| Middleton | 6 | 350 | 2.1 | 207 | 150 | 9.8% |
| Yarmouth | 6 | 350 | 2.1 | 196 | 142 | 9.3% |
| Port Hastings | 6 | 350 | 2.1 | 189 | 137 | 9.0% |
| Chester Bas | 6 | 350 | 2.1 | 182 | 132 | 8.7% |
| Barrington | 9 | 150 | 1.4 | 127 | 93 | 9.4% |
| Liverpool | 4 | 350 | 1.4 | 110 | 80 | 7.9% |
| Weymouth | 4 | 350 | 1.4 | 106 | 77 | 7.5% |
| Musq. Hbr | 3 | 350 | 1.1 | 95 | 69 | 9.0% |
| Annapolis Royal | 3 | 350 | 1.1 | 93 | 67 | 8.9% |
| Whycoco. | 3 | 350 | 1.1 | 87 | 63 | 8.3% |
| S. Brookfield | 3 | 350 | 1.1 | 78 | 56 | 7.4% |
| St. Peter's | 3 | 350 | 1.1 | 77 | 56 | 7.3% |
| Shelburne | 3 | 350 | 1.1 | 75 | 54 | 7.1% |
| Tatamagouche | 3 | 350 | 1.1 | 71 | 51 | 6.7% |
| Baddeck | 5 | 150 | 0.8 | 69 | 50 | 9.1% |
| Cooks Cove | 3 | 350 | 1.1 | 46 | 34 | 4.4% |
| Parrsboro | 3 | 350 | 1.1 | 46 | 34 | 4.4% |
| Margaree Forks | 3 | 350 | 1.1 | 37 | 27 | 3.5% |
| Mabou | 3 | 150 | 0.5 | 36 | 26 | 8.0% |
| Sheet Hbr. | 3 | 350 | 1.1 | 35 | 26 | 3.4% |
| Carlton | 3 | 350 | 1.1 | 33 | 24 | 3.1% |
| Cape North | 3 | 150 | 0.5 | 31 | 23 | 6.9% |
| Wreck Cove | 3 | 150 | 0.5 | 30 | 22 | 6.7% |
| New Germany | 3 | 150 | 0.5 | 29 | 21 | 6.5% |
| Cheticamp | 3 | 150 | 0.5 | 24 | 18 | 5.4% |
| S. Lochaber | 3 | 150 | 0.5 | 24 | 17 | 5.2% |
| U. Musq. | 3 | 350 | 1.1 | 20 | 14 | 1.9% |
| Liscomb | 3 | 150 | 0.5 | 10 | 7 | 2.1% |
| Framboise, CB | 3 | 150 | 0.5 | 9 | 7 | 2.0% |
| **Total** | **318** | | **104** | **9246** | **6712** | **8.9%** |

## 5. Conclusions

A network of fast charging hubs enables EV adoption by alleviating range and charge anxiety, and allows vehicle owners to travel as they expect, and to any destination. We present a two-step method relying on common publicly available data to design such a network that provides equitable access, is appropriately sized, and is reliable. We argue

that while detailed Lagrangian (vehicle tracking) datasets may produce high accuracy over a limited geographic domain, existing daily Eulerian (location specific) traffic data are more suitable for jurisdiction-wide infrastructure planning.

The first step of the method deals with the realities of the electrical and road infrastructure, which change slowly through time. The second step is applied separately to each hub location identified in the first step, and addresses the expected travel patterns of EVs and its regional variability. Consequently, step 1 (site selection) is insensitive to technology, and should provide confident guidance for both the near term and long. In contrast step 2 (hub sizing) scales with technology, population growth, travel and vehicle technology and is highly sensitive to these assumptions. This difference implies that site expansion in site layout, buried conduit capacity, etc., be considered in all planning for the hub sizing.

The method is built around the principle that coordinated central planning is needed for efficient investment, and the concept of a charging hub, a location with numerous individual DCFCs, providing reliability, redundancy, and convenience for users. Hubs also realize economies of scale in construction, and offer a venue for small businesses to cater to drivers while their vehicles charge. The principal design choice within the method is the target hub spacing, which must be related to the single-charge range of EVs in 'worst case' scenarios. The network of hub locations is produced using a 'seed' location and iterative growth method, constrained by this target hub spacing and infrastructural restriction of electrical supply and major route junctions defining hub-eligible locations.

In a case study for Nova Scotia, Canada, the presented method was used to produce a maps of hub locations and DCFC counts at each. This map portrays a snapshot of necessary charging infrastructure for 2030, when EVs are projected to make up 15% of the vehicle fleet. The network consists of 280 DCFCs of 350 kW each at 35 hubs that meet what we have termed 'rigorous' geographic criteria, and an additional 38 DCFCs of 150 kW, at 10 hub sites that require 'loosened criteria' and fill in charging deserts.

**Author Contributions:** Conceptualization, E.B., S.B., L.H., B.P. and J.A.; methodology, N.S.P., L.G.S., E.B., S.B., L.H., B.P. and J.A.; software, N.S.P. and L.H.; validation, N.S.P., L.G.S., E.B. and J.A.; formal analysis, N.S.P., L.H. and E.B.; investigation, N.S.P., L.G.S. and E.B.; resources, L.G.S. and B.P.; data curation, L.G.S. and L.H.; writing—original draft preparation, N.S.P.; writing—review and editing, N.S.P. and L.G.S.; visualization, N.S.P. and L.H.; supervision, S.B. and B.P.; project administration, L.G.S., S.B. and J.A.; funding acquisition, S.B. and B.P. All authors have read and agreed to the published version of the manuscript.

**Funding:** This research was conducted with funding support from the Nova Scotia Department of Natural Resources and Renewables.

**Data Availability Statement:** All data sources not shared with the authors under non-disclosure agreement are online and are cited in the text.

**Conflicts of Interest:** There are no conflict of interest. The funders had no role in the design of the study; in the collection, analyses, or interpretation of data; in the writing of the manuscript; or in the decision to publish the results.

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
