# Peer review of "Regional Electric Vehicle Fast Charging Network Design Using Common Public Data"

_wevj, doi:10.3390/wevj13110212_

Round 1

Reviewer 1 Report

Please find attached file.

Author Response

Thank you for the thoughtful review. Please see the attached document.

Reviewer 2 Report

The authors prepared a manuscript about two-stage design strategy for a network of charging hubs. The authors` design strategy based on daily traffic volume data. The study is in scope of WEVJ and can be useful for the researchers who work on developing networks for electric vehicle charging. Specifically, using daily traffic volume data makes this manuscript useful for the research community. The novelty of the paper is clearly explained in the first part of the paper, also the article is written well. Therefore, the article can be accepted in the present form.

Author Response

Thank you for the timely review. Please see the attached document.
